# The Effects of Action Observation Therapy as a Rehabilitation Tool in Parkinson’s Disease Patients: A Systematic Review

**DOI:** 10.3390/ijerph19063311

**Published:** 2022-03-11

**Authors:** Ioannis Giannakopoulos, Panagiota Karanika, Charalambos Papaxanthis, Panagiotis Tsaklis

**Affiliations:** 1Biomechanics and Ergonomics Laboratory, Department of Physical Education and Sports Science (DPESS), University of Thessaly, 42100 Trikala, Greece; ergomechlab@uth.gr (I.G.); pkaranik@uth.gr (P.K.); recherche@chu-dijon.fr (C.P.); 2L’Unité Mixte de Recherche (UMR) INSERM 1093 CAPS (Cognition, Action et Plasticité Sensorimotrice), Université Bourgogne Franche-Comté, UFR des Sciences du Sport, F-21000 Dijon, France; 3Pôle Recherche et Santé Publique, CHU Dijon Bourgogne, F-21000 Dijon, France; 4Department of Molecular Medicine and Surgery, Growth and Metabolism, Karolinska Institute, 17164 Solna, Sweden

**Keywords:** Parkinson’s disease, neurological rehabilitation, mirror neurons, action observation, motor control, systematic review

## Abstract

During Action Observation (AO), patients observe human movements that they then try to imitate physically. Until now, few studies have investigated the effectiveness of it in Parkinson’s disease (PD). However, due to the diversity of interventions, it is unclear how the dose and characteristics can affect its efficiency. We investigated the AO protocols used in PD, by discussing the intervention features and the outcome measures in relation to their efficacy. A search was conducted through MEDLINE, Scopus, Cochrane, and WoS until November 2021, for RCTs with AO interventions. Participant’s characteristics, treatment features, outcome measures, and main results were extracted from each study. Results were gathered into a quantitative synthesis (MD and 95% CI) for each time point. Seven studies were included in the review, with 227 participants and a mean PEDro score of 6.7. These studies reported positive effects of AO in PD patients, mainly on walking ability and typical motor signs of PD like freezing of gait. However, disagreements among authors exist, mainly due to the heterogeneity of the intervention features. In overall, AO improves functional abilities and motor control in PD patients, with the intervention dose and the characteristics of the stimulus playing a decisive role in its efficacy.

## 1. Introduction

Action observation (AO) therapy/training is based on the significant discovery of mirror neurons, initially found in the monkey cerebral cortex [1,2]. Precisely, it was observed that these neurons discharge, not only during the execution of goal-directed actions, but also during the observation of the same actions performed by other macaques. The brain areas containing mirror neurons constitute the Mirror Neuron System (MNS). The MNS was found to be also present in the human brain [3,4]. For example, during AO, the excitability of the motor cortex is enhanced [5], and brain areas in the frontal and parietal lobes are recruited, similarly to motor execution [6]. In addition, the MNS is involved in ‘’imitation’’ within a circuit, engaging the inferior parietal lobule, the inferior frontal gyrus, and the premotor cortex [7]. Linking all the above, it is derived that the MNS is significantly exploited in humans during AO training [8]. During an AO session, patients carefully observe movements performed by an actor, which in some cases they try to imitate physically later. Inside our brain, we map the representation of what we see onto motor systems, gaining knowledge of those actions by executing them internally [9]. From that concept, it has been widely demonstrated that the link between observation and action can promote motor learning [10].

AO training has been long studied in healthy people, but only recently made its appearance in clinical practice. Several studies have shown that it is effective as a way to learn a new motor skill or enhance its performance in healthy individuals, in an analogous manner to physical exercise [11,12,13,14]. In rehabilitation, an adequate number of studies have been published so far that demonstrate positive effects of AO training [8,15,16,17]. Bearing in mind only neurological cases, stroke is undoubtedly the most common one where AO has been tested and systematically reviewed, with the majority of studies reporting a clear benefit of it [7,18,19,20,21,22,23,24,25,26,27]. Unlike stroke, AO is less investigated in Parkinson’s Disease (PD). Although the presence of the MNS is known, it is still unclear if and how it works in these people through AO training. PD patients are accompanied by a deficit in the cortical network subserving movement preparation, which is subsequently translated into symptoms like bradykinesia and akinesia [28]. However, AO can be used as peripheral feedback and sensorimotor integration by modulating cortical plasticity and may provide them with convenient cues to enhance their motor function. Despite the small number of studies, this fact can be supported [29,30,31]. However, due to the diversity of interventions in these studies, it is unclear how the stimuli and the dose of interventions affected the outcomes. Also, the measures used relied on their evaluations on scales that measure specific aspects of the disease, like the PDQ-39, making us consider that they may be restricted [32]. Identification of the most appropriate characteristics of an AO protocol, as well as the most suitable outcome measures, could enhance the efficacy of AO in the clinical world.

The above background drives us to conduct a systematic review investigating the different experimental protocols of AO training used in PD patients so far, by discussing the features of each intervention (stimulus and dose) in relation to their efficacy and the outcome measures used in relation to their reliability and compatibility, to identify the most appropriate treatment and experimental design for this disease.

## 2. Materials and Methods

The present systematic review was conducted and written in accordance with the guidelines outlined by the updated Preferred Reporting Items for Systematic Reviews and Meta-Analysis (PRISMA) 2020 statement [33].

### 2.1. Information Sources and Search Strategy

To identify studies, a comprehensive search was conducted through the following databases: MEDLINE via PubMed, Scopus, Cochrane Library, and Elsevier via Science Direct until November 2021. A similar search strategy in all databases’ titles and abstracts was carried out with the terms “Action observation”, “Parkinson’s disease”, “Parkinson”, and other different synonyms and expressions. In addition, a manual search in each article’s reference list was made to identify additional studies. The detailed version of search strategy is provided as Appendix A.

### 2.2. Eligibility Criteria

The studies included in the review met the following inclusion criteria: (1) male or female participants with a clinical diagnosis of Parkinson’s disease in consonance with the UK Parkinson’s Disease Society Brain Bank criteria [34]; (2) studies of randomized controlled trials (RTCs) where an AO intervention was implemented with no restrictions on its features; (3) comparison with other intervention, or placebo, or no intervention; (4) outcome measures related on motor and functional recovery conducted at any time point; and (5) studies published in English. The following exclusion criteria were considered: (1) simultaneous interventions, (2) studies where only brain imaging methods were used as outcome measures, and (3) pilot studies.

### 2.3. Study Selection Process

Two researchers searched independently through the literature and at first, all titles and abstracts from each search were screened to identify relevant studies. After duplicates and irrelevant records were removed, several relevant reports were sought for retrieval and assessed for eligibility through full-text screening by the two reviewers, independently. Subsequently, they excluded any reports with reasons, before ending up with the final studies of the review. In case of disagreement at any stage of the process, a third reviewer facilitated the final decision process. All studies were imported, screened, and assessed through the EndNote^x9^ software (Clarivate, Philadelphia, PA, USA).

### 2.4. Data Items and Collection Process

The extracted data of the final studies included the characteristics of participants, the features of interventions, the outcome measures, and the main findings. The first reviewer sought and extracted these data at first, and afterwards, the second reviewer checked the correctness of the process, and in case of disagreement, a third reviewer made the final decision. Reviewers worked independently during the process.

### 2.5. Synthesis Methods

Results of included studies were gathered into a quantitative synthesis and presented as tables for outcome measures at all time points as mean difference and 95% confidence interval.

### 2.6. Risk of Bias Assessment

The PEDro scale was used for the assessment of the risk of bias among the included studies. PEDro scale is a valid measure of methodological quality of clinical trials in rehabilitation [35]. It consists of 11 items including external validity (item 1), internal validity (items 2–9), statistical reporting (items 10 and 11), and each one of them contributes 1 point to a total score of 10, except from one dichotomous item (yes/no). PEDro scale is considered to meet interval level measurement, allowing score comparisons between studies [36]. Scores of: <4 are considered ‘poor’, 4–5 are considered ‘fair’, 6–8 are considered ‘good’, and 9–10 are considered ‘excellent’ [37]. Two reviewers assessed the risk of bias in each study independently and a third reviewer was recruited in case of any argument.

## 3. Results

### 3.1. Study Selection

Overall, 254 records were identified through all database searches. After 170 duplicates had been removed, titles and abstracts from the remaining 169 records were screened and the full text of 31 was assessed for eligibility. Finally, seven studies were included in this review. A comprehensive flowchart diagram of the study selection process is presented in Figure 1.

### 3.2. Study Characteristics

Characteristics of the included studies are presented in Table 1.

#### 3.2.1. Participants

Studies recruited at least 18 and a maximum of 64 PD patients (≥18 and ≤64) with mild to moderate disease severity (Hoehn & Yahr score < 4) and were eligible if they were able to walk unassisted. Also, all participants had a disease duration of at least 9 years except from one study, in which they had a disease duration of at least 5 years [38]. Moreover, the authors considered as eligible a Mini-Mental Status Examination score of above 24, which indicates an absence of dementia. Two of the included studies also recruited healthy controls in addition to PD patients [30,38]. Lastly, five studies included patients with freezing of gait, with at least one episode per week lasting at least 2 s [38,39,40,41,42].

**Table 1 ijerph-19-03311-t001:** Characteristics of the included studies.

Study	Participants’ Characteristics	AO	Control	Design/Dose	Task/Stimulus	Outcome Measures
**Pelosin et al., 2010** [42]	**PD patients: (*n* = 18)**, FOG-Q item 3 ≥ 2 and item 4 ≥ 1, MMSE > 24 **AO group: (*n* = 9)**, 68.8 ± 4.1 years, F:M 7/6 **Control group: (*n* = 9)**, 70.2 ± 6.8 years, F:M 6/4	Watched videos of movements and strategies to circumvent FoG episodes and then practiced the observed actions	Watched sequences of static pictures of landscapes and then practiced the same actions as the experimental group	**Sessions:** 3 per week **Session duration:** 1 h **Protocol duration:** 4 weeks **Total sessions:** 12	**Movements:** weight shifting, step, turn around chair, step over obstacle, walk straight-through doorway **Perspectives:** 3rd person—frontal	FOG-Q, FoG-diary, TUG, 10M-WT, BBS, Tinetti scale and PDQ-39 **Time points:** baseline; 2 days after; 1, 2, 3, and 4 weeks after
**Pelosin et al., 2013** [30]	**PD patients: (*n* =20)**, H&Y: 1–3, MMSE ≥ 24 **Healthy patients:** **(*n* = 14)** **AO group: *n* =10 (PD)**, 68.8 ± 7.4 years, F:M 3/7, DD: 9.1 ± 3.7, UPDRS: 18.9 ± 4.2 ***n* = 7 (H)**, 64.3 ± 8.6 years, F:M 3/4 **Control group: *n* = 10 (PD)**, 66.4 ± 8.9 years, F:M 6/4, DD: 8.9 ± 3.1, UPDRS: 19.2 ± 5.4 ***n* = 7 (H), 69.2 ± 9.6 years, F:M 4/3**	Watched videos of repetitive finger movements	Listened acoustic cues	**Sessions:** 1 **Duration:** 6 min	**Movements:** opposition of right thumb to all other fingers at 3 HZ pace **Perspectives:** 3rd person	**Primary:** SMR of self-paced finger movements **Secondary:** Inter-tapping interval and touch duration (kinematic parameters) **Time points:** Baseline, after, 45′ after, 2 days after
**Jaywant et al., 2016** [43]	**PD patients: (*n* = 23)**, H&Y 1–3, UPDRS gait item ≥ 1 **AO group: (*n* = 12)**, 63.7 ± 6.2 years, DD: 11.6 ± 4.9 years **Control group: (*n* = 10)**, 70.2 ± 6.8 years, DD: 9.5 ± 3.7 years	Watched videos of walking trials and judged whether the observed action was a PD or healthy pattern	Watched videos of water moving roughly and calmly and judged whether the motion of the water was rough or calm	**Sessions:** 1 per day **Session duration:** not specified **Protocol duration:** 1 week **Total sessions:** 7	**Movements:** walking in hallway **Perspectives:** 3rd person—frontal, lateral, and posterior views	PDQ-39 and stride frequency, number-duration of walking periods during straight walking, walking with turns, and dual task walking **Time points:** baseline, 1 day after
**Agosta et al., 2017** [38]	**PD patients:****(*n* =25)**, H&Y < 4, MMSE > 24, FOG-Q item 3 ≥ 2, DD ≥ 5 years **Healthy patients:** **(*n* = 19)**, 66 ± 8 years, F:M 10/9 **AO group: *n* =12 (PD)**, 69 ± 8 years, F:M 2/10 **Control group: *n* = 13 (PD), 64 ± 7 years, F:M 5/8**	Watched videos of movements with the help of auditory cues and then imitated them at the same beats	Watched videos of static landscape images and then executed the same movements as the experimental group	**Sessions:** 3 per week **Session duration:** 1 h (24 min observation—36 min action) **Protocol duration:** 4 weeks **Total sessions:** 12	**Movements:** weight shifting, stepping forward-backward-side, turn around chair, step over obstacle, walk straight-through doorway **Perspectives:** 3rd person—frontal view	UPDRS III (on/off), H&Y (on/off), FOG-Q, UPDRS II-FoG (on/off), PDQ-39, BBS, 10M-WT **Time points:** baseline, after (week 4), after 1 month (week 8)
**Mezzarobba et al., 2017** [40]	**PD patients: (*n* = 22)**, FoG, H&Y 1–3, BDI ≤ 16, MMSE > 24 **AO group: (*n* = 12)**, 74.6 ± 5.9 years, F:M 5/7, DD: 10.7 ± 3.44 years **Control group: (*n* = 10)**, 72 ± 5.8 years, F:M 3/7, DD: 9.4 ± 4.8 years	Watched videos of gait-related gestures and after video clip practiced the same observed action for the same amount of time (x2)	The same motor gestures performed in the same order and time by means of visual (floor) or auditory (metronome) cues	**Sessions:** 2 per week **Session duration:** 1 h **Protocol duration:** 8 weeks **Total sessions:** 16	**Movements:** weight shifting + step, gait initiation, turn around, step over obstacle, STW, walk straight- through doorway **Perspectives:** 3rd person—frontal-lateral views	**Primary:** NFOGQ (duration & severity) **Secondary:** UPDRS II, III, H&Y, PDQ-39, 6M-WT, BBS, TUG, improvement index **Time points:** baseline, after, 1 month after, 3 months after
**Pelosin et al., 2018** [39]	**PD patients: (*n* = 64)**, FOG-Q: item 2 ≥ 1 & item 4 ≥ 2, H&Y 2–3, MMSE > 24, unassisted walk **AO group:** **(*n* = 33)**, 70.4 ± 4.5 years, F:M 17/16, DD: 10.7 ± 3.9 years **Control group:** **(*n* = 31)**, 72.8 ± 3.1 years, F:M 16/15, DD: 9.5 ± 4.2 years	Watched videos of functional movements and then practiced the observed actions with the help of physiotherapist	Watched videos of static landscape images and then practiced the same actions as the experimental group	Sessions: 2 per week Session duration: 45 min Protocol duration: 5 weeks Total sessions: 10	**Movements:** weight shifting, weight shifting + step, turn around chair, step over obstacle, walk straight-through doorway **Perspective:** 3rd person—frontal view	**Primary:** FOG-Q **Secondary:** TUG, 10M-WT, BBS **Time points:** baseline, 1 week after training, 4 weeks after training
**Mezzarobba et al., 2020** [41]	**PD patients: (*n* = 22)**, FoG, H&Y 1–3, BDI ≤ 16, MMSE > 24 **AO group: (*n* = 12)**, 74.6 ± 5.9 years, F:M 5/7, DD: 10.7 ± 3.44 years **Control group: (*n* = 10)**, 72 ± 5.8 years, F:M 3/7, DD: 9.4 ± 4.8 years	Watched videos of gait-related gestures and after video clip practiced the same observed action for the same amount of time (x2)	The same motor gestures performed in the same order and time by means of visual (floor) or auditory (metronome) cues	**Sessions:** 2 per week **Session duration:** 1 h **Protocol duration:** 8 weeks **Total sessions:** 16	**Movements:** weight shifting + step, gait initiation, turn around, step over obstacle, STW, walk straight—through doorway **Perspectives:** 3rd person—frontal—lateral views	STW time, COM’s & COP’s time—position, **Task:** STW **Time events:** initiation, flexion phase, extension phase, unloading phase, and stance phase **Time points:** baseline, after, 1 month after, 3 months after

AO, Action Observation; PD, Parkinson’s disease; F, female; M, male; DD, disease duration; FoG, Freezing of Gait; FOG-Q, Freezing of Gait questionnaire; FoG-diary, Freezing of Gait diary; H&Y, Hoehn & Yahr; MMSE, Mini-Mental Status Examination; UPDRS, Unified Parkinson’s Disease Rating Scale; PDQ-39, Parkinson’s Disease Questionnaire-39 items; 10M-WT, 10 Meters Walking Test; BBS, Berg Balance Scale; SMR, Spontaneous Movement Rate; NFOGQ, New Freezing of Gait Questionnaire; 6M-WT, 6 Minutes Walking Test; TUG, Timed Up and Go test.; STW, Sit To Walk.

#### 3.2.2. Action Observation Interventions

All the included studies implemented AO interventions alone with the help of a laptop used to project the video-clips. Only Pelosin et al. [39] used a wall in front of participants for the projection. Every session was composed of an observation and an action phase, except from two studies in which there was no action phase after observation [30,43]. In the first one [30], patients watched videos of repetitive finger movements and were only instructed to concentrate on how the actions were performed, while in the second one [43], they observed walking trials and then were asked to judge if the observed task was performed by a healthy or a parkinsonian model. Three studies asked patients to imitate the observed tasks after the observation phase [38,39,42], while in the remaining two studies, after observation of each video, patients were asked to imitate the action of that video, while still being in the observation phase [40,41]. The duration of each session ranged between 45–60 min, while Jaywant et al. did not specify the exact duration [43]. There was also a single session study with a session duration of 6 min [30]. As far as the motor contents of the stimulus are concerned, the majority of studies used gait-related tasks such as walking, stepping, obstacle avoidance etc. in their videos [38,39,40,41,42,43]. One study only used intransitive upper limb tasks like repetitive finger movements [30]. Three of the included studies, in addition to visual stimuli, implemented auditory cues (metronome), which were associated with the movements [38,40,41]. All studies used third-person perspective from frontal [38,39,42], frontal-lateral [40,41], and frontal-lateral-posterior [43] views. Pelosin et al. [30] also used a third-person perspective, however, they did not specify the exact view from which videos were presented. Models that executed these tasks were healthy individuals in the majority of the studies [30,38,39,40,41,42], while Jaywant et al. used also PD patients in addition to healthy models [43].

#### 3.2.3. Control Interventions

In three of the included studies, during the observation phase, control groups watched videos of static pictures and landscapes as a placebo, while during the action phase they practiced the same actions as the experimental groups following the instructions of an operator [38,39,42]. Two of the studies did not implement an observation phase in their control interventions and during the action phase, subjects performed the same motor gestures as the AO group in the same order and time using visual or auditory cues [40,41]. Finally, in the two studies that there was no action phase after observation in the experimental groups, subjects in the control groups either listened to acoustic cues [30] or watched videos of moving water [43]. In the latter, after observation, participants judged whether the motion of the water was rough or calm. In all studies, control groups received the same dose of intervention as the experimental groups, concerning the duration and frequency of the protocols.

#### 3.2.4. Outcome Measures and Time Points

For the quality of life (QoL) assessment, the Parkinson’s Disease Questionnaire-39 (PDQ-39) was used in four studies [38,40,42,43], while disease severity was assessed with the Unified Parkinson’s Disease Rating Scale (UPDRS) in two studies [38,40]. Four studies assessed the freezing of gait episodes among the patients with the Freezing of Gait Questionnaire (FoG-Q) [38,39,42], the New Freezing of Gait Questionnaire (NFoG-Q) [40], and the Freezing of Gait diary (FoG-diary) [42]. Also, many functional measures were conducted like the 10 Meters Walking Test (10M-WT) [38,39,42], the 6 Minutes Walking Test (6M-WT) [40], the Timed Up and Go test (TUG) [39,40,42], the Berg Balance Scale (BBS) [38,39,40,42], and the Tinetti Scale [42], for the assessment of gait speed, gait endurance, balance, and functional mobility. Furthermore, Jaywant et al. [43] measured and analyzed spatial-temporal gait parameters like stride frequency and number and duration of walking periods during straight walking, walking with turns, and dual-task walking; while Pelosin et al. [30] measured spontaneous movements rate and kinematic parameters such as inter-tapping interval and touch duration, during self-paced finger movements. Finally, Mezzarobba et al. [41] determined the position and time of the Centre of Mass (COM) and the Centre of Pressure (COP) during a sit to walk task at different time events (initiation, flexion phase, extension phase, unloading phase, and stance phase). As far as the time points of the measures are concerned, all studies conducted measurements before the start of the intervention and after the completion of it. However, four of them assessed patients immediately after the end of the intervention [30,38,40,41], while the other three conducted measures 1 day after [43], 2 days after [42], and 1 week after the training period [39]. For the long-term adaptation, patients were assessed after 45′ and 2 days in the single session study [30], 1 month [38,39,40,41,42], and 3 months after the end of the intervention [40,41].

### 3.3. Risk of Bias in Studies

The included studies in this systematic review presented PEDro scores that ranged between 5 and 8 points with an average score of 6.7 points. Five of them had good scores and high methodological quality [38,40,41,42,43], while the remaining two [30,39] presented fair scores and moderate methodological quality. More specifically, all studies did not report blinding of participants and therapists and two of them also did not report blinding of assessors [39,43]. Three of them had no allocation concealment [30,39,42], three did not acknowledge for intention-to-treat analysis [30,38,39], and one study did not specify the number of missing data at follow-up [30]. The PEDro score of each included study is presented in Table 2.

### 3.4. Results of Included Studies

The results of the included studies generally suggest that AO training is effective in PD patients, both in motor and functional-related measures and self-reported questionnaires. However, significant arguments exist among the authors, especially in between-group comparisons. The results of each study are presented in Table 3. Agosta et al. [38] found that immediately after the end of AO training (W4), participants reduced FoG severity and enhanced UPDRS-III ON, PDQ-39, BBS, and 10M-WT scores, with improvements maintained until the follow up (W8), while the UPDRS-III ON score showed a trend towards a significant improvement at W8 in the AO group compared to the control group. Similar results for the AO group (AO plus sonification) were presented by Mezzarobba et al. [40] with significant positive effects in FoG severity and duration (NFoG-Q, primary outcome measure), PDQ-39 subscales, UPDRS-III, UPDRS-II, and BBS, while the control group (cue protocol) did not show any relevant gain effects in these scales. Positive effects of AO training on FoG were also reported in the other two studies [39,42]. In the first [42], the mean FoG-Q score was significantly reduced both at the end of training and at the follow-up examination, with the scores at follow-up being significantly lower in the AO than in the control group. Also, FoG episodes were significantly reduced in the AO group during all follow-up testings (POST, W1, W2, W3, W4), with the between-groups difference being significant at W2, W3, and W4 (largest effect). Motor performance and quality of life tests (TUG, 10M-WT, Tinetti scale, and BBS) showed no difference between groups at any point. In the second study [39], FoG-Q scores followed a similar pattern over time. They significantly improved in both groups post-treatment, but the improvement was retained up to the follow-up evaluation only in the AO group. Similarly, BBS and TUG improved in both groups at post-treatment assessment, but the positive effect at follow-up, retained only in the AO group. Finally, walking assessment (10M-WT) showed no significant difference between groups at any time point. Jaywant et al. [43] on the other hand, despite reporting a significant change in the AO group on post-training PDQ-39-mobility scores, found no significant difference between the groups. In addition, no difference was reported between the groups in any of the walk-related outcome measures (walking speed, stride length-frequency, swing time, and gait asymmetry) after training. Mezzarobba et al. [41] compared an AO group with a control cue group during a sit-to-walk task by collecting biomechanical data (COM, COP and moving timings). Despite that there was no significant difference on the sit-to-walk times at any time point after training, the COP profiles showed significant differences between the two groups post-training, with patients in the AO enhancing the quality of COP profiles even 3 months post-intervention. At last, regarding the single session experiment for bradykinesia during repetitive finger movements [30], a reduction of the inter-tapping interval was found for the AO (post, 45′, 2-days), when compared to the control intervention, as well as an increase in self-paced movement rate, 45′ and 2-days post-intervention.

## 4. Discussion

The scope of the present systematic review was to outline the features of AO interventions in relation to their efficacy in PD patients and the outcome measures used in relation to their reliability and compatibility. The inclusion criteria were met by seven studies including 194 PD participants with mild to moderate disease severity (Hoehn & Yahr score < 4), disease duration of at least 5 years, and dementia absence, as well as 33 healthy individuals. These studies focused on the effects of AO interventions on PD related motor symptoms, like freezing of gait and bradykinesia, quality of life, and functional mobility. As said, FoG is a gait-related symptom in PD, which increases the risk of fall and decreases the QoL [44,45]. Studies have shown that during walking, PD patients display a decrease in supplementary motor area activity, which is compensated by increased recruitment of basal ganglia [46]. However, during demanding movements like obstacle avoidance, this subcortical hyperactivity fails and the phenomenon of FoG may occur. AO training can be used to reduce FoG symptoms, as it boosts the recruitment of the MNS and frontal-parietal areas which, during sudden changes in the interaction between body and environment, are responsible for conscientious mechanisms, reducing the possibility of FoG [47]. Indeed, four studies that investigated the efficacy of AO training on FoG, showed not only positive short-term, but also long-term effects [38,39,40,42]. Also, the AO interventions among studies, induced positive changes in clinical outcomes such as walking ability, balance, QoL, and PD-related scales, as well as upper limb function after a single-session intervention. This is explained by the fact that through AO training, the individual can reorganize the neural circuits that connect the motor cortex with basal ganglia and the projections from the motor cortex to the thalamus [8,48]. Lastly, although Agosta et al. [38] found no significant differences between groups, within-group improvements were found in the AO group and fMRI measures revealed an interrelation of it, with increased recruitment of the frontal-parietal MNS during observation and execution of a task.

Two of the studies [30,39] presented a PEDro score of 5 points, which indicates a moderate methodological quality, in contrast with the other five which scored 7 or above and had good quality. These two studies did not apply concealed allocation and intention to treat analysis, facts that may overestimate or underestimate the effects of AO. Additionally, except for one study [39] with 64 participants, the sample size was relatively small among studies and ranged from 18 to 25 subjects. Studies with a larger sample size are needed to reliably investigate the effects of AO training on PD patients.

### 4.1. Dose/Design of the Interventions

Noteworthy differences can be found in the duration and frequency of the interventions among the included studies, which may influence their efficacy. Most of them, reported a treatment duration between 4 and 8 weeks, a sufficient period for both short and long-term detectable changes in PD patients. Oppositely, Jaywant et al. [43] implemented a daily protocol that lasted 1 week. Although seven sessions seem to be an adequate number for positive changes, duration of only 1 week may not be enough to display the efficacy of AO. Also, high-frequency training with five sessions per week may not be effective in these patients [49]. PD patients seem to need low to intermediate-frequency protocols with two to three sessions per week, to maximize the retention of the acquired motor skills. This guideline was followed from the rest five studies, while Pelosin et al. [30] investigated the effects of a single session protocol.

Session duration is also a feature that must be considered when designing an AO treatment for PD. Although an optimal duration is not suggested, most studies conducted AO sessions that lasted on average 1 h, a duration that seems to be adequate. Only Jaywant et al. [43] did not specify the exact duration and Pelosin et al. [30] conducted a single session experiment that lasted 6 min.

### 4.2. Characteristics of the Stimuli/Task

The video-clips used to represent motor tasks are the main interaction between the patient and the treatment. Sarasso et al. [50] suggested that the heterogeneity in these videos can influence the efficacy of AO treatment. The type of movements to be used is a major factor. The motor repertoire is a concept used to describe the previous acquisition of motor skills by an individual. When the observed actions belong to the motor repertoire of the individual, the MNS activity is greater and so are the adaptations [51,52]. This drives us to consider that it may be better to deliver PD patients; video-clips representing models with the same pathological condition. However, finding subjects imitating pathological patterns, or patients with the same pathological movement patterns is not easy, because neurological conditions have a big diversity of motor symptoms. All studies in this review used healthy subjects as models, except for Jaywant et al. [43], in which PD patients were also included.

The presence or absence of an object characterizes the actions as transitive (with object interaction) and intransitive (without object interaction), respectively. Transitive actions have been shown to have increased MNS activity in neuroimaging studies [7,53], while intransitive ones activate the MNS in a more restrictive manner [32,54]. The included studies used both kinds within the same treatment, thus it is difficult to compare their efficacy. However, the minor activation of the MNS during intransitive actions does not necessarily mean that an AO treatment based on these actions has less benefit than one with transitive. While transitive is more complex, patients with neurological disorders that have attention deficits or less capacity to follow the action for a longer time may become cognitively overloaded [32]. On the other hand, the imitation of simple intransitive actions would be effortless for them. Consequently, it is safe to assume two things. Firstly, future studies should investigate the effects of AO using only intransitive actions, because although they lead to less brain activity, PD patients can be more focused on the kinematics of the action. Secondly, the combination of the two kinds could enhance the efficacy of AO, only when there is a progression in complexity of actions, starting from simple-intransitive to challenging-transitive, paying attention to the level of impairment of each patient. Two studies implemented this progression [40,42].

Evidence also exists that the person-related perspective from which the motor task is observed (first or third-person) may also influence the efficacy of the treatment. It is proved that the mirror neurons in the premotor cortex of the macaques are view-dependent and are activated when the movement is observed from a specific perspective [55,56]. Depending on it, the observation of a movement activates different MNS regions [57]. This also applies to humans. More specifically, there is higher MNS activation and easiness in imitation of an action observed in a first compared to a third-person perspective [58,59]. Furthermore, the kinesthetic perception of the observer with a first-person perspective is enhanced, facilitating the vividness of internal representation, thus improving the ability of physical execution [59,60,61]. The included studies used only a third-person perspective. Future studies should focus on different person-related perspectives to establish the best possible one for PD patients. On top of that, when using a third-person perspective, viewing perspective is also a characteristic that could influence the efficacy of AO. A view that highlights all the elements of the observed motor task is more probable to enhance motor learning. Actions taking place along the frontal plane, like side-stepping, should be recorded from a frontal or posterior view, while movements along the sagittal (walking,) should be recorded from a lateral view. Six studies presented the stimulus from frontal or posterior views [38,39,40,41,42,43], with three of them using also lateral views [40,41,43].

Mezzarobba et al. [40,41] implemented an auditory cue in combination with a standard AO protocol, and they reported magnified effects. An explanation could be that the overall cognitive load might be reduced [62]. A two-sensory congruent stimulus can help patients improve their attention, thus enhancing their performance. Also, fMRI studies have shown that this kind of stimuli increases the functional connectivity between basal ganglia and frontal-parietal cortical areas, which are important in cognitive process and sensory integration [62]. However, more studies are needed to further investigate the effects of this stimulation.

### 4.3. Outcomes Measures

Except for the dose of the intervention and the characteristics of the stimulus, the outcome measures are also important when evaluating the efficacy of AO. The scales or outcome measures used in the application of AO in PD, base their evaluations mostly on scales that measure specific aspects of the disease, like the PDQ-39 questionnaire. While these scales are vital for understanding the effects of the treatment in this neurological disease, the broad implications of AO (motor and cognitive implications) make us consider that they may be restricted [32]. The evaluations of AO training should be based on more functional scales in terms of functional recovery. In fact, the included studies used a combination between clinical scales (PDQ-39, UPDRS III, etc.) and functional measures (10M-WT, TUG, BBS, etc.), with FoG and walking ability being the most assessed variables among them. As it seems, evaluating as many clinical and functional aspects as possible is a safe approach to detect any changes.

### 4.4. Limitations

The limitations of the present systematic review must be highlighted to safely conclude about the efficacy of AO training. First of all, the small number of participants in the majority of studies, as well as the small number of included studies, may affect the outcomes. Also, the variance in the features of AO protocols and the diversity in the time points of the assessments was high. Only two studies conducted long-term assessments, making it hard to conclude about the long-term efficacy of the treatment. Lastly, reporting of the results was inadequate in some studies.

## 5. Conclusions

To conclude, AO training is a treatment that improves functional mobility, motor control, and clinical aspects in PD patients. However, the training dose and the visual stimulus characteristics play a major role in the effectiveness of interventions. Despite there being an agreement among studies about using stimulus with transitive actions, future studies should also incorporate and investigate the use of intransitive actions. Also, studies that compare the effects of different kinds of visual perspectives (person and viewing perspective) and model types (healthy vs. PD), as well as studies with large sample sizes, would give a clearer view of the most appropriate AO protocols for these patients.

## Figures and Tables

**Figure 1 ijerph-19-03311-f001:**
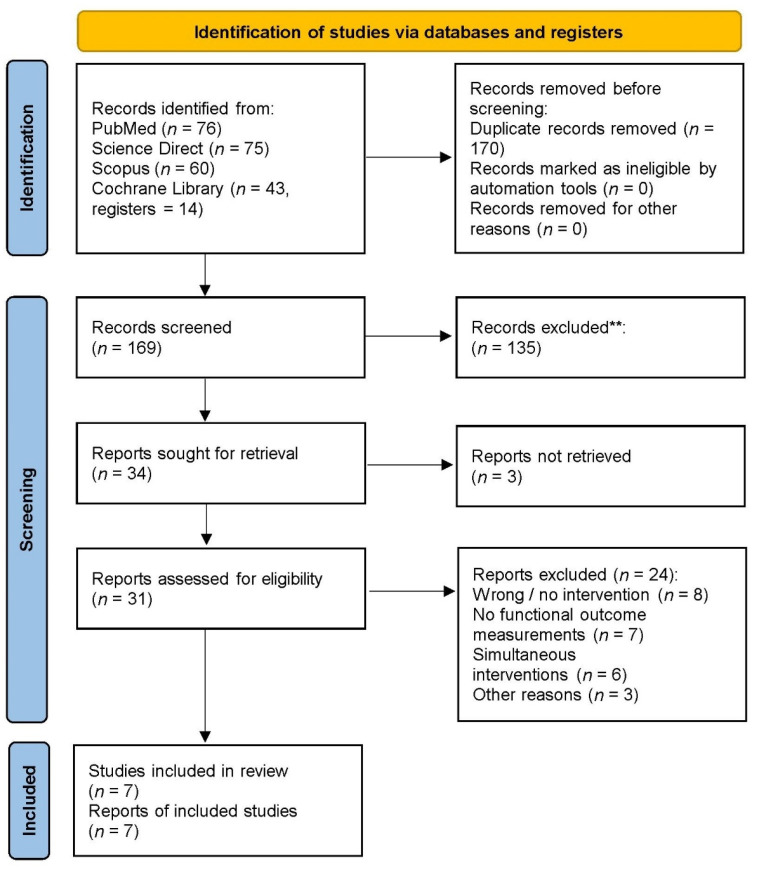
Flowchart diagram of the study selection process. **, irrelevant records excluded.

**Table 2 ijerph-19-03311-t002:** Risk of bias in the included studies.

	Eligibility Criteria & Source	Random Allocation	Concealed Allocation	Baseline Comparability	Blinding of Participants	Blinding of Therapists	Blinding of Assessors	Adequate Follow-Up (>85%)	Intention-to-Treat Analysis	Between-Group Statistical Comparisons	Reporting of Point Measures & Variability	Total Score (0–10)
**Pelosin et al., 2013** [30]	yes	1	0	1	0	0	1	0	0	1	1	**5**
**Agosta et al., 2017** [38]	yes	1	1	1	0	0	1	1	0	1	1	**7**
**Pelosin et al., 2018** [39]	yes	1	0	1	0	0	0	1	0	1	1	**5**
**Mezzarobba et al., 2017** [40]	yes	1	1	1	0	0	1	1	1	1	1	**8**
**Mezzarobba et al., 2020** [41]	yes	1	1	1	0	0	1	1	1	1	1	**8**
**Pelosin et al., 2010** [42]	yes	1	0	1	0	0	1	1	1	1	1	**7**
**Jaywant at al., 2016** [43]	yes	1	1	1	0	0	0	1	1	1	1	**7**

**Table 3 ijerph-19-03311-t003:** Results of included studies.

Outcome Measures		Time Points	Experimental Group	Control Group	Mean Difference [95% CI]
**Pelosin et al., 2010** [42]					
Action Observation Training Group (Experimental Group) vs. Landscape Observation Training Group (Control Group)					
FoG-Q		Post 2 Days	12.8 (2.0)	14.4 (1.9)	−1.6 [−3.40. 0.20]
		Post 4 Weeks	14.1 (2.8)	16.4 (2.5)	**−2.3 [−4.75, 0.15]**
TUG, 10M-WT, Tinetti Scale, BBS and PDQ-39		Post 2 Days	Not significant		
		Post 1 Week	Not significant		
		Post 2 Weeks	Not significant		
		Post 3 Weeks	Not significant		
		Post 4 Weeks	Not significant		
FoG-diary (total number of episodes)		Post 2 Days	Not significant		
		Post 1 Week	Not significant		
		Post 2 Weeks	** *p* ** ** < 0.05**		
		Post 3 Weeks	** *p* ** ** < 0.05**		
		Post 4 Weeks	** *p* ** ** < 0.05**		
**Outcome Measures**		**Time Points**	**Between** **Groups Difference**		
**Pelosin et al., 2013** [30]					
Action Observation Training Group (Experimental Group) vs. Acoustic Training Group (Control Group)					
Self-paced Movement Rate		Post	Not significant		
		Post 45′	** *p* ** ** = 0.007**		
		Post 2 Days	** *p* ** ** = 0.004**		
Inter-tapping Interval		Post	** *p* ** ** = 0.019**		
		Post 45′	** *p* ** ** < 0.001**		
		Post 2 Days	** *p* ** ** < 0.001**		
Touch Duration		Post	Not significant		
		Post 45′	Not significant		
		Post 2 Days	Not significant		
**Outcome Measures**		**Time Points**	**Experimental Group**	**Control Group**	**Mean Difference [95% CI]**
**Jaywant et al., 2016** [43]					
Action Observation Training Group (Experimental Group) vs. Landscape Observation Training Group (Control Group)					
PDQ-39		Follow-up (1 week)	n/a	n/a	3.08 [−2.97, 9.12]
Straight Line Walking	Walking Speed (m/s)	Follow-up (1 week)	1.19 (0.15)	1.18 (0.08)	0.01 [−0.32, 0.34]
	Stride Length (m)	Follow-up (1 week)	1.35 (0.21)	1.34 (0.12)	0.01 [−0.46, 0.48]
	Stride Frequency (strides/s)	Follow-up (1 week)	0.89 (0.06)	0.89 (0.06)	0.00 [−0.17, 0.17]
	Swing Time (% of stride)	Follow-up (1 week)	45.6 (1.6)	44.8 (1.7)	0.80 [−3.78, 5.38]
	Gait Asymmetry	Follow-up (1 week)	0.03 (0.02)	0.02 (0.01)	0.01 [−0.03, 0.05]
Walking with Turns	Walking Speed (m/s)	Follow-up (1 week)	1.19 (0.13)	1.19 (0.08)	0.00 [−0.30, 0.30]
	Stride Length (m)	Follow-up (1 week)	1.36 (0.20)	1.35 (0.11)	0.01 [−0.44, 0.46]
	Stride Frequency (strides/s)	Follow-up (1 week)	0.89 (0.07)	0.88 (0.06)	0.01 [−0.17, 0.19]
	Swing Time (% of stride)	Follow-up (1 week)	45.3 (1.3)	44.7 (1.6)	0.60 [−3.44, 4.64]
	Gait Asymmetry	Follow-up (1 week)	0.03 (0.01)	0.03 (0.01)	0.00 [−0.03, 0.03]
Dual Task Walking	Walking Speed (m/s)	Follow-up (1 week)	1.17 (0.18)	1.17 (0.15)	0.00 [−0.46, 0.46]
	Stride Length (m)	Follow-up (1 week)	1.34 (0.23)	1.34 (0.14)	0.00 [−0.53, 0.53]
	Stride Frequency (strides/s)	Follow-up (1 week)	0.88 (0.07)	0.88 (0.08)	0.00 [−0.21, 0.21]
	Swing Time (% of stride)	Follow-up (1 week)	45.3 (1.7)	44.6 (1.9)	0.70 [−4.30, 5.70]
	Gait Asymmetry	Follow-up (1 week)	0.03 (0.03)	0.03 (0.02)	0.00 [−0.07, 0.07]
**Outcome Measures**		**Time Points**	**Experimental Group**	**Control Group**	**Mean Difference [95% CI]**
**Agosta et al., 2017** [38]					
Action Observation Training Group (Experimental Group) vs. Landscape Observation Training Group (Control Group)					
H&Y-off		Post (W4)	2.5 (0.5)	2.3 ± 0.4	0.20 [−0.17, 0.57]
H&Y-on		Post (W4)	2.4 (0.4)	2.2 ± 0.3	0.20 [−0.09, 0.491]
		Post (W8)	2.2 (0.4)	2.2 ± 0.4	0.00 [−0.33, 0.33]
UPDRS-III-off		Post (W4)	35.0 (10.9)	33.8 ± 9.0	1.20 [−6.89, 9.29]
UPDRS-III-on		Post (W4)	23.3 (7.8)	24.2 ± 8.3	**−1.10** ** [−7.55, 5.35]**
		Post (W8)	23.3 (10.1)	22.1 ± 8.4	**1.20** ** [−6.55, 8.95]**
FoG-Q		Post (W4)	9.7 (3.4)	10.9 ± 3.0	−1.20 [−3.79, 1.39]
		Post (W8)	10.2 (2.4)	11.3 ± 3.0	−1.10 [−3.31, 1.11]
UPDRS-II-FoG-off		Post (W4)	1.64 (0.94)	1.92 ± 0.79	−0.28 [−0.98, 0.42]
		Post (W8)	2.13 (0.99)	2.0 ± 1.1	0.13 [−0.73, 0.99]
UPDRS-II-FoG-on		Post (W4)	1.18 (0.87)	1.25 ± 0.75	−0.07 [−0.73, 0.59]
		Post (W8)	0.89 (0.93)	0.92 ± 0.95	−0.03 [−0.80, 0.74]
PDQ-39		Post (W4)	19.0 (9.2)	14.0 ± 8.9	−0.07 [−0.73, 0.59]
		Post (W8)	17.0 (7.0)	16.7 ± 10.5	−0.03 [−0.80, 0.74]
BBS		Post (W4)	53.6 (2.6)	54.4 ± 2.4	−0.80 [−2.82, 1.22]
		Post (W8)	53.4 (2.7)	54.4 ± 2.2	−1.00 [−3.06, 1.06]
10 M-WT-normal (s)		Post (W4)	8.2 (1.1)	7.2 ± 1.2	1.00 [0.08, 1.92]
		Post (W8)	8.2 (1.4)	7.68 ± 1.7	0.52 [−0.75, 1.79]
10 M-WT-fast (s)		Post (W4)	6.0 (1.4)	5.6 ± 1.0	0.40 [−0.59, 1.39]
		Post (W8)	6.1 (2.0)	6.0 ± 1.6	0.00 [−1.51, 1.51]
**Outcome Measures**		**Time Points**	**Between Groups Difference**		
**Mezzarobba et al., 2017** [40]					
Action Observation plus Sonification Training Group (Experimental Group) vs. Motor Gesture with Visual & Auditory Cues Training Group (Control Group)					
NFoG-Q		Post	** *p* ** ** ≤ 0.001**		
		Post 1 Month	** *p* ** ** ≤ 0.001**		
		Post 3 Months	** *p* ** ** ≤ 0.001**		
PDQ-39 mobility		Post	** *p* ** ** ≤ 0.05**		
		Post 1 Month	** *p* ** ** ≤ 0.001**		
		Post 3 Months	** *p* ** ** ≤ 0.001**		
UPDRS-III		Post	** *p* ** ** ≤ 0.001**		
		Post 1 Month	** *p* ** ** ≤ 0.05**		
		Post 3 Months	** *p* ** ** ≤ 0.05**		
PDQ-39-bodily discomfort		Post	** *p* ** ** ≤ 0.001**		
		Post 1 Month	** *p* ** ** ≤ 0.05**		
		Post 3 Months	** *p* ** ** ≤ 0.05**		
PDQ-39-Total		Post	Not significant		
		Post 1 Month	** *p* ** ** ≤ 0.01**		
		Post 3 Months	** *p* ** ** ≤ 0.01**		
UPDRS-II		Post	Not significant		
		Post 1 Month Post 3 Months	** *p* ** ** ≤ 0.05**		
			** *p* ** ** ≤ 0.01**		
BBS		Post	Not significant		
		Post 1 Month	** *p* ** ** ≤ 0.05**		
		Post 3 Months	Not significant		
6MWT		Post	Not significant		
		Post 1 Month	Not significant		
		Post 3 Months	** *p* ** ** ≤ 0.05**		
TUG		Post	Not significant		
		Post 1 Month	Not significant		
		Post 3 Months	Not significant		
MPAS		Post	Not significant		
		Post 1 Month	Not significant		
		Post 3 Months	Not significant		
PDQ-39 cognitions		Post	Not significant		
		Post 1 Month	Not significant		
		Post 3 Months	Not significant		
**Outcome Measures**		**Time Points**	**Experimental Group**	**Control Group**	**Mean Difference** **[95% CI]**
**Pelosin et al., 2018** [39]					
Action Observation Training Group (Experimental Group) vs. Landscape Observation Training Group (Control Group)					
FoG-Q		Post 1 Week Post 4 Weeks	9.7 (5.8) 9.4 (5.7)	10.5 (4.8) 12.0 (5.7)	−0.8 [−3.47, 1.87] **−2.6 [−5.46, 0.26]**
TUG		Post 1 Week Post 4 Weeks	12.2 (4.9) 12.9 (4.1)	13.4 (6.1) 15.5 (6.8)	−1.2 [−3.98, 1.58] **−2.6 [−5.43, 0.23]**
BBS		Post 1 Week Post 4 Weeks	51.3 (5.7) 51.5 (5.5)	52.4 (4.5) 49.6 (5.7)	−1.1 [−3.67, 1.47] **1.9 [−0.91, 4.71]**
10M-WT		Post 1 Week Post 4 Weeks	10.7 (3.9) 12.3 (4.3)	12.9 (4.3) 13.9 (5.4)	−2.2 [−4.26, −0.14] −1.6 [−4.05, 0.85]
**Outcome Measures**		**Time Points**	**Between Groups Difference**		
**Mezzarobba et al., 2020** [41]					
Action Observation plus Sonification Training Group (Experimental Group) vs. Motor Gesture with Visual & Auditory Cues Training Group (Control Group)					
Sit-to-walk times (s)		Post Post 1 Month Post 3 Months	Not significant Not significant Not significant		
COP Profiles		Post Post 1 Month Post 3 Months	**Significant difference (30–50% range) ** **Significant difference (40–50% range) ** **Significant difference (14–50% range)**		

H&Y, Hoehn & Yahr; FoG-Q, Freezing of Gait Questionnaire; UPDRS, Unified Parkinson’s Disease Rating Scale; PDQ-39, Parkinson’s Disease Questionnaire-39 items; 10M-WT, 10 m Walking Test; BBS, Berg Balance Scale; NFoG-Q, New Freezing of Gait Questionnaire; 6M-WT, 6 Minutes Walking Test; FoG-diary, Freezing of Gait diary; TUG, Timed Up and Go test; COP, Centre of Pressure; W, week. Significant results are reported in bold.

## Data Availability

The data presented in this study are available on request from the corresponding author.

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
