# Peer review of "The Effects of Action Observation Therapy as a Rehabilitation Tool in Parkinson’s Disease Patients: A Systematic Review"

_ijerph, 2022, doi:10.3390/ijerph19063311_

Round 1

Reviewer 1 Report

The abstract should include a brief description of Action Observation in PD as the lector could know what the authors are talking about.

The text is cumbersome and sometimes arduous to read, especially the results. A somewhat more friendly way of conveying the content of the article could be tried that engages readers more and makes it more attractive.

English needs to be improved.

Reviewer 2 Report

Giannakopoulos, et al. reviewed efficacy of AO in PD patients. The authors comprehensively addressed literatures and also assessed risk of bias for included studies. I agree with the authors’ point that AO improves motor symptoms/signs of PD. The authors’ insights in Discussion sound. However, I think that the paper contains some problems for publication as a systematic review.

  1. Too many studies are excluded during the study selection process. The authors excluded 135 out of 169 records. I wonder, why reports (documents) of so many records were not available. The flowchart of Figure 1 has asterisks (**) in this process, but it is not explained what this means.

  1. Study groups / authors are often overlapped among included studies. The same authors / groups tend to similar conclusions. In this context, it is important how and why so many studies were excluded, as discussed in the query 1.

  1. Although Table 1 addresses demography of participants, the contents can be more simplified and unified into a single table layout. I recommend making a concise table displaying sample size (F/M, AO/non-AO), ages, median disease duration, mean H&Y stage, and UPDRS of the total participants from the included studies. Successful examples of unified layouts of tables are easily (open-accessed) available from published systematic reviews. For example, Zhu SZ, et al. Front Aging Neurosci 2020; 12: 592212.

  1. Table 3 essentially contains seven tables and hardly readable, which may arise from inconsistent format among cited studies. Number of columns differs, ranging from three to five, among studies. I recommend attempt to simplify parameters to unify the table format, as discussed in the query 3. I also feel that Results of Included Studies in the page 8 could be more systematically and concisely written if Table 3 is improved.

  1. In Table 3, statistical significance is displayed in different ways among the seven studies: ‘p-values’, ‘95%-CI’, and ‘significant difference (% range)’. Why is this? Original paper by Agosta F (ref. 38) displayed p-value, whereas Table 3 of present study described 95%-CI.

  1. Table 3 mainly addresses difference between AO-group and non-AO group at each timepoint. By contrast, cited seven studies often emphasize longitudinal improvement of PD-motor signs/symptoms between before and after AO. I recommend displaying comparisons among timepoints (significance in improvement of clinical scores after AO); although settings of timepoints may differ among studies, it can be detailed in caption of the table. Not only intergroup differences but also longitudinal differences in each group are important. For example, FOG-Q score in Agosta’s study (ref. 38) did not differ between AO-group and Control group at W4 or W8, but the study reported that FOG-Q significantly improved after AO in AO-group. Although I know the authors to address this point in Result and Discussion, the current layout of Table 3 does not have power to emphasize this fact.

  1. Abstract told ‘these studies reported positive effects of AO in PD patients, mainly on walking ability and typical motor signs of PD’. Please clarify what ‘typical motor signs’ mean in Abstract. I feel a strength of AO to be efficacy against frozen gait, as reported in cited studies.

  1. When the authors cite literatures/evidences in the text, please clarify whether studies are based on human-subject or animal models. For example, ‘a fMRI study on healthy people reported…’, ‘tracer experiments on macaques have described…’, or ‘basic studies using PD-patient-derived iPS-neuron indicated…’

  1. Overall, this systematic review does not answer clinical questions proposed in Introduction; which stimuli and how many dose are recommended for AO therapy in PD patients. Of course, it partially arises from limited number of reports and participants, as authors noted in Limitation. More importantly, AO-protocols and settings of outcome measurements varied among studies. I suggest attempt to set up protocol/outcome parameters that are consistently and commonly applicable toward heterogeneous studies. Generally, quantitative data, such as a FOG-Q or PDQ-39 score, is hardly manipulated because timepoints vary among studies. However, qualitative parameters, for example, ‘presence /absence of efficacy to frozen gait’, can be applied to various studies. In conclusion, current version is suitable for a review article narratively introducing AO-therapy in PD patients, however, I feel that some improvement to clearly answer clinical questions is required to publish it as an original research paper.

Reviewer 3 Report

3 AO is not a tool

14 Its dose?

16 Elsevier? … Pubmed, Scopus and WoS

19? gathered into a quantitative synthesis (MD and 95% CI)?
A draft of pico and eligibility is missing in the methods .. what outcomes were included?

30 Start the intro with a few lines on parkinson's and the motor and cognitive deficits it can cause

Argue the need for rehabilitation for these people:

“In addition to motor symptoms, cognitive dysfunctions very often occur in patients with PD, in fact once the cognitive impairment can be objectified in patients with PD, it is called Mild Cognitive Impairment in PD (PD-MCI) [1]. Recent research suggests that cognitive function can be improved, or stabilized, through cognitive training. And although the ideal type or frequency of intervention of cognitive training therapy for PD-MCI is not clear to date, several interventions have also been proposed that combine cognitive and physical training [2]. Therefore, exergames show great potential for PD rehabilitation since the use of video games requires the user to perform physical movements while performing cognitive exercises [3]. Teaching motor strategies, without cues might be an intriguing innovative approach to rehabilitation, that matters most on appropriate allocation of attention, and lightening cognitive load [4]. In this scenario, one of these strategies, such as observation of action (AO), is developing interesting perspectives in the rehabilitation projects of people with PD.”

(ref1: https://doi.org/10.1186/s12984-019-0492-1 , erf2: https://doi.org/10.1177%2F0891988718807973 , ref3: https://pubmed.ncbi.nlm.nih.gov/32478581/ , ref 4: https://doi.org/10.1002/gps.4845 )

 Search strategy first,

Pedro indirectly certifies the risk of bias, so I think it is appropriate to support RoB2 for example

148 “design” instead of “dose”. Task-orientation instead of Stimulus

307 Change dose .. a sort of “AO design”

324 as above

Round 2

Reviewer 2 Report

I have no further comments on this paper. Minor points have been revised well.